# The Transcriptomic Analysis of the Response of *Pinus massoniana* to Drought Stress and a Functional Study on the ERF1 Transcription Factor

**DOI:** 10.3390/ijms241311103

**Published:** 2023-07-05

**Authors:** Jinfeng Zhang, Dengbao Wang, Peizhen Chen, Chi Zhang, Sheng Yao, Qingqing Hao, Romaric Hippolyte Agassin, Kongshu Ji

**Affiliations:** State Key Laboratory of Tree Genetics and Breeding, Key Laboratory of Forestry Genetics & Biotechnology of Ministry of Education, Co-Innovation Center for Sustainable Forestry in Southern China, Nanjing Forestry University, Nanjing 210037, China; zhangjinfeng1012@126.com (J.Z.); dbw@njfu.edu.cn (D.W.); pei_jane@126.com (P.C.); chiz@njfu.edu.cn (C.Z.); yaosheng0827@126.com (S.Y.); hqq@njfu.edu.cn (Q.H.); hippolyteagassin@gmail.com (R.H.A.)

**Keywords:** *Pinus massoniana* Lamb., drought stress, transcriptome sequencing, transcription factor, *AP2/ERF*

## Abstract

*Pinus massoniana* is a major fast-growing timber tree species planted in arid areas of south China, which has a certain drought-resistant ability. However, severe drought and long-term water shortage limit its normal growth and development. Therefore, in this study, physiological indices, and the transcriptome sequencing and cloning of *AP2/ERF* transcription factor of *P. massonsiana* were determined to clarify its molecular mechanism of drought stress. The results showed that stomatal conductance (Gs) content was significantly decreased, and superoxide dismutase (SOD) activity, and malondialdehyde (MDA) and abscisic acid (ABA) content were significantly increased under drought stress. Transcriptomic analysis revealed that compared to the control, 9, 3550, and 4142 unigenes with differential expression were identified by comparing plants subjected to light, moderate or severe drought. *AP2/ERF* with high expression was screened out for cloning. To investigate the biological functions of *ERF1*, it was over-expressed in wild-type *Populus davdianaand* × *P. bolleana* via the leaf disc method. Under drought stress, compared to wild-type plants, *ERF1* over-expressing poplar lines (OE) maintained a higher photosynthetic rate and growth, while the transpiration rate and stomatal conductance significantly decreased and water use efficiency was improved, indicating that drought tolerance was enhanced. This study provides an insight into the molecular mechanism of drought stress adaptation in *P. massoniana.*

## 1. Introduction

China is one of the drought-sensitive areas; climate predictions have forecasted that hot and dry events will become more frequent in the coming decades [1,2]. Drought has seriously affected agricultural and forestry production in arid and semi-arid areas in China. In order to resist drought stress, plants adapt to a drought environment mainly through morphological, physiological, biochemical and cellular steady-state changes, such as leaf rolling, stomatal closure, membrane stability, osmotic adjustment, antioxidant accumulation, reactive oxygen species (ROS) scavenging and transcription activation, which are affected by drought [3]. These complex drought-resistant systems are variable among plant species. Molecular responses to drought stress have been extensively studied in broad leaf species, but studies in conifers are limited [4]. It is of great significance to explore the adaptability of conifers to drought, to reveal the mechanism of drought resistance and cultivate more drought-resistant plants.

The abscisic acid (ABA)-mediated signaling pathway has the core effects on plant abiotic stress [5]. Drought may trigger the production of ABA. The accumulated ABA binds to the PYR/PYL/RCAR receptors to form dimers that bind to protein phosphatases 2C (PP2Cs). PP2C releases sucrose non-hydrolytic protein kinases 2 (SNF1-related protein kinases 2), phosphorylates the corresponding transcription factors, regulates the expression of ABA-responsive genes, causes stomatal closure, and improves the drought resistance of plants [6]. In addition to the ABA signal transduction pathway, gibberellin (GA), jasmonic acid (JA), ethylene (ET) and other plant hormone signal transduction pathways also play a vital role in the plant response to drought stress. Transcription factors make a critical difference in activating or restraining the conveyance of defense-related genes in these signal transduction pathways. In order to improve plant stress resistance and better adapt to stress environment, plants can control the interaction between different signaling pathways through transcription factors. Some members of the *AP2/ERF* transcription factors are involved in these pathways [7]. The *AP2/ERF* (APETALA2/ERF) transcription factor (TFs) family is one of the largest families of plant TFs, and has drawn more and more attention from researchers. The *ERF* subfamily is an important *AP2/ERF* family of TFs closely related to abiotic stress. Current studies have shown that the *ERF1* gene in *Arabidopsis thaliana* can join in JA, ET, and ABA signal transduction pathways and activate the expression of stress resistance genes. *A. thaliana* over-expressed with *ERF1* significantly increased drought resistance, and the transgenic plants could reduce leaf water loss by reducing stomatal pore size [8]. *DREB* subfamily TFs are a class of abiotic stress-related TFs. The *DREB* subfamily can specifically bind DRE/CRT cis-acting elements, activate downstream stress resistance gene expression, and improve plant resistance to drought stress independent of the ABA signaling transcription pathway [9]. *Malus sieversii* (Led.) Roem overexpression with *MsDREB6.2* would affect stomata and the roots of plants, thus enhancing drought resistance [10]. In conclusion, a variety of metabolic pathways are involved in the plant defense against abiotic stresses. Regulating the expression of related genes can improve the adaptability of plants to adverse environments. In recent years, a lot of research has been conducted on the molecular mechanism of drought stress [8,11,12], but little is conducted on conifers.

*P. massoniana* is the main local afforestation tree species in southern China and the pioneer tree species for improving the environment and greening barren mountains. It has excellent characteristics such as rapid growth, a high yield and barrenness resistance. It plays an irreplaceable role in the production of solid wood and resin [13]. The wide distribution environment of *P. massoniana* is complex, and the uneven temporal and spatial distribution of annual rainfall limits the large-scale expansion of its plantation. Researchers have studied the response of *P. massoniana* seedlings from different provenances [14] and families [15] to drought, which provides a certain theoretical basis and reference basis for the screening and evaluation of drought-tolerant germplasms, but there are still some limitations to the selection of drought resistance indicators. Transcription analysis has been used to study gene regulation under drought conditions and to identify many genes involved in the drought response [16]. Although some progress has been made, the regulation of the plant drought response remains to be further studied. In this study, we analyzed the photosynthetic, physiological and transcriptomic responses of *P. massoniana* under drought conditions, and confirmed the crux transcription factors involved in the drought response. *AP2/ERF* with high expression was screened from transcriptome data for cloning. Using *Agrobactrium tumefaciens*-mediated transformation, *ERF1* was transferred into *Populus davidiana* × *P. bolleana* to determine if the transgenic poplar could improve drought tolerance. This study will provide a valuable molecular reference for improving the drought resistance of *P. massoniana*. The result can also help understand the drought resistance mechanisms and screening drought resistance factors in the future.

## 2. Results

### 2.1. Effects of Drought Stress on Seedlings of P. massoniana

Drought stress can affect the growth of *P. massoniana* seedlings seriously (Figure 1). Under drought stress, the seedlings grew slowly and wilted. With the aggravation of drought stress, the stomatal conductance (Gs) of *P. massoniana* seedlings decreased rapidly (Figure 2A). In order to adapt to the arid environment, superoxide dismutase activity (SOD) and the content of malondialdehyde (MDA) increased gradually (Figure 2C,D). The content of ABA increased (Figure 2B), and was 3.8 times higher than that of the control under severe stress.

### 2.2. Transcriptome Sequencing and De Novo Assembly and Annotation

Briefly, 162.04 gb clean data were obtained from the transcriptome sequencing of 12 samples. After assembly, the uni-gene function annotation was performed including a comparison with NR (NCBI, non-redundant protein sequences), Swiss-Prot (a manually annotated and reviewed protein sequence database), KEGG, COG (Clusters of Orthologous Groups of proteins), KOG (EuKaryotic Orthologous Groups), and GO and Pfam (protein family) databases, and 35,398 uni-gene annotation results were obtained. Of the 35,398 uni-genes, 11,967 (33.80%) and 19,432 (54.89%) had significant match in the COG and GO databases, respectively. In addition, 13,113 (37%), 20,085 (56.75%) and 30,908 (87.31%) uni-genes had annotations in the KEGG, KOG, and eggNOG databases, respectively. As shown in Figure 3, the GO classification included biological process (25,976 genes), cellular component (19,839 genes), and molecular function (26,525 genes) (Figure 3A). The COG classification included 25 functional categories, including translation, ribosomal structure and biogenesis (1341 genes, 11.31%), posttranslational modification, protein turnover, chaperones (1209 genes, 10.19%), carbohydrate transport and metabolism (1186 genes, 10%) (Figure 3B). Furthermore, the annotated genes were enriched in 131 KEGG pathways, including those for plant hormone signal transduction, ribosome, plant–pathogen interaction, RNA transport, plant circadian rhythm and photosynthesis.

To identify DEGs between different samples, we first assessed gene expression levels based on the threshold of fragments per kb of transcript per million mapped reads (FPKM), using Cufflinks (v2.2.1) to measure the gene expression distribution in each sample (Appendix A). The Wilhelm Wien map showed the specific genes (272, 255, 246 and 584 expressed genes in G1, G2, G3 and G4, respectively) and shared genes (14 and 402 expressed genes) (Appendix A). We used Pearson product-moment correlation coefficient as an indicator to assess the correlation between biological repeats (Appendix A). Then, DEGs were identified using an FC of ≥ 2 and an FDR of < 0.01 as the criteria in each pairwise comparison. A total of 7701 DEGs were identified, including 4080 up-regulated genes and 3621 down-regulated genes. The DEGs were annotated using the COG (2537 DEGs, 38.96%), GO (3384 DEGs, 51.97%), KEGG (2223 DEGs, 34.14%), KOG (3058 DEGs, 46.97%), Pfam (5144 DEGs, 79.00%), Swiss-Prot (4825 DEGs, 74.11%), eggNOG (5846 DEGs, 89.83%), and NR (6459 DEGs, 99.20%) databases. The distribution of up-regulated and down-regulated genes and their FCs is clearly shown in each paired comparison of MA to volcano maps (Appendix A).

### 2.3. Identification of DEGs in Drought-Treated P. massoniana

In order to investigate molecular activity under different drought conditions, we analyzed the differential expression of three treatments (LD, MD and SD). Among them, 9, 3550 and 4142 genes were differentially expressed between LD and CK, MD and CK, and SD and CK, respectively. Of these, 1 up-regulated and 8 down-regulated were detected for LD and CK; 1872 up-regulated and 1678 down-regulated geners were detected for MD and CK; 2207 up-regulated and 1935 down-regulated genes were detected for SD and CK.

The annotated DEGs in MD vs. those in CK were analyzed. A total of 1112 (36.34%) annotated DEGs were identified, including 518 up-annotated and 594 down-annotated KEGG paths, and 20 minimum Q values were selected to draw the path enrichment scatter plot. The up-regulated genes were functionally allocated to 50 biological genes including carbon metabolism (26 DEGs, 8.78%), starch and sucrose metabolism (24 DEGs, 8.88%), phenylpropanoid biosynthesis (22 DEGs, 7.43%), photosynthesis (19 DEGs, 6.42%), and amino acid biosynthesis (17 DEGs, 5.74%). Based on the path enrichment factor analysis, fimonene and pinene degradation, and photosynthesis, flavone and flavonol biosynthesis were significantly enriched. The down-regulated genes were classified as carbon metabolism (29 DEGs, 9.12%), the biosynthesis of amino acids (25 DEGs, 7.89%), starch and sucrose metabolism (20 DEGs, 6.29%), and amino sugar and amino tide metabolism (19 DEGs, 5.97%).

In addition, the annotated DEGs in SD vs. CK were also analyzed. A total of 1181 (34.28%) annotated DEGs were identified, including 525 up-regulated genes and 656 down-regulated genes (Figure 4A,B). The up-regulated genes were classified as carbon metabolism (26 DEGs, 8.58%), starch and sucrose metabolism (22 DEGs, 7.26%), phenylpropanoid biosynthesis (20 DEGs, 6.60%), and carbon fixation in photosynthetic organisms (18 DEGs, 5.94%) (Figure 4C,D). The enrichment factor of limonene and pinene degradation was more significant than that of other approaches. The down-regulated genes were classified as genes associated with carbon metabolism (33 DEGs, 9.48%), the biosynthesis of amino acids (28 DEGs, 8.05%) and glycolysis/gluconeogenesis (23 DEGs, 6.61%) (Figure 4E,F).

### 2.4. RT-qPCR Validation

Using qRT-PCR analysis to verify the dependability of RNA-Seq, we randomly chose nine DEGs showing significant up- or down-regulation in the RNA-seq data, including *SPI2*, *NPF*, *MYB*, *WRKY*, *CPS1*, *bZIP*, *ERF1*, *ASR* and AUX/IAA (Table 1). The expression pattern was validated by relative expression (RT- qPCR) and FPKM (RNA-seq) values. *SPI2*, *NPF*, *MYB, WRKY* and *CPS1* uni-genes showed up-regulated expression under MD and SD drought stress (Figure 5A–E). Three uni-genes, *ERF1, ASR* and *AUX/IAA*, were up-regulated under LD and down-regulated under MD and SD (Figure 5G–I). One uni-gene, *bZIP*, showed a down-regulated manifestation with increasing drought stress (Figure 5F). Although the expression of MYB genes is different, the trends of other genes were similar. Therefore, the transcriptome sequencing results were valid.

### 2.5. Expression Pattern Analysis of PmERF1

*AP2/ERF* (c62205.graph_c0) genes with relatively high expression levels were screened from transcriptome data (SRA accession: PRJNA595650) under drought stress for cloning. The ORF of this gene was 798 bp and it encoded 265 amino acids and one termination codon. BlastP analysis of the AP2/ERF protein in *P. massoniana* showed high sequence similarity with the *P. mugo* 1B protein and that it was relatively close to that of *Actinidia chinensis var.* 1B. However, it is relatively far from that of *Brassica campestris* L., *Glycine max* (Linn.) Merr., and *A. thaliana*, which are more highly evolved, indicating that this amino acid sequence is highly conserved in the AP2 domain (Figure 6A) and belongs to the ERF family. Other names for ERF1B are ERF1 and ethylene response factor 1 according to TariH software analysis; therefore, we named it PmERF1. Its molecular formula, molecular weight and theoretical iso-electric point (PI) were C_1319_H_2118_N_388_O_409_S_11_, 30.31 kDa and 6.37, respectively, as determined via ExPasy ProtParam analysis. Moreover, PmERF1 has 34 negatively charged residues (Asp + Glu) and 31 positively charged residues (Arg + Lys). The instability index (II) was 59.69, indicating that the protein was an unstable protein. Its fat index was 76.35 and the average hydrophilicity (GRAVY) was −0.392; therefore, the PmERF1 protein was a hydrophilic protein. In order to determine the subcellular localization of ERF1, transient expression vectors of green fluorescent proteins (GFPs) (35S::*PmERF1*-GFP and 35S::-GFP (as a negative control)) were transferred into tobacco (*Nicotiana benthamiana*) leaves. The fluorescence signal of the 35S::*PmERF1*-GFP protein was observed at the nucleus, whereas the 35S::-GFP protein was distributed without specific localization (Figure 6B). These results indicated that PmERF1 was specifically localized in the nucleus.

Gene expression pattern is closely related to gene function. The tissue-specific expression pattern of *PmERF1* in different organs of *P. massoniana* (young needles, old needles, xylem, young stems, old stems and roots) was detected via RT-qPCR. *PmERF1* transcription levels were expressed in all organs and tissues, but the expression level was highest in old needles, while the expression level was weak in the xylem (Figure 6C). To understand the response of *PmERF1* to drought stress, the expression level of *PmERF1* under dehydration and ABA was tested using qRT-PCR (Figure 6D). ABA is a hormone that plays an important role in the drought response, and it mediates drought response genes. Therefore, the response of *PmERF1* to ABA treatment was tested. After ABA treatment, the expression of *ERF1* was up-regulated and then it decreased, increased about 2.5 times after 3 h, and then sharply decreased after 12 h (Figure 6E).

### 2.6. Drought Tolerance of OE Lines under Drought Stress

At present, there is no established genetic transformation system in *P. massoniana*. Therefore, the pSuper::*Pm*ERF1 vector was transformed into woody wild-type (WT) *Populus davidiana* × *P. bolleana* using the leaf disc method, and then *PmERF1* over-expressing transgenic poplar (OE) lines were generated (Appendix A). Transgenic lines were confirmed via PCR using gene-specific primers and RT-qPCR, all of which exhibited the expected band (Figure 7A). The expression levels of OE-2 and OE-8 were the highest in different transgenic *PmERF1* lines (Figure 7B). We selected them as experimental materials for physiological and drought stress experiments. 

In order to study the drought resistance of OE lines, drought treatment was applied with water interruption for 7 days. As can be seen from the figure, the leaves of WT strains were seriously wilted, while those of the OE strains were still swollen (Figure 7C,D). With the progress of drought treatment, the net photosynthetic rate (Pn) of WT plants decreased significantly and showed almost no photosynthetic activity on the seventh day, while the Pn of OE lines decreased significantly only on the fourth day, then slowed down gradually and remained at a certain level on the seventh day, indicating that the OE lines could still produce the energy material needed for growth through photosynthesis under drought conditions (Figure 7F). Under drought treatment, Gs and Tr rates in WT and OE lines decreased and showed a consistent trend of decline, but the decline in OE lines was slower than that in the WT (Figure 7G,H). In addition, the leaf relative water content (RWC) of OE lines was higher than that of the WT plants (Figure 7E). We also studied leaf water loss in the WT and OE lines. Under dehydration conditions, the WT lines lost water faster than the OE plants did (Figure 7I). In conclusion, these results showed that under drought stress, OE plants were more tolerant than were the those of the WT lines.

## 3. Discussion

Drought stress is one of the major abiotic stresses in the world, which seriously affects the biochemical and physiological process of plants. We found that under drought stress, plants often deal with the damage caused by increasing the growth of underground parts and reducing the growth of above ground parts, so as to reduce the pressure of osmotic stress placed on normal physiological activities. MD and SD plants grew slowly under drought stress, and the height and diameter of LD plants were 3.3 times of those of SD plants [17]. Drought stress caused the stomatal closure of plants, severely reducing their photosynthetic activity [18]. In this study, the Gs value of *P. massoniana* seedlings decreased rapidly with the intensification of drought stress. This indicated that drought would cause stomatal opening to decrease or even close, weaken transpiration, affect CO_2_ absorption, and decrease the photosynthetic rate. This phenomenon is consistent with the research results of Song et al. [19] for the photosynthetic characteristics of bamboo willow. Plants used antioxidant defense systems such as enzymes (SOD, CAT, POD, APX and GPX) and other compounds to control and eliminate excess ROS to enhance drought tolerance [20], in order to enhance the osmotic adjustment and the cell water absorption ability that maintains the life activity of the seedlings. With the intensification of drought stress, the SOD activity and MDA content of *P. massoniana* seedlings increased. This indicated that drought stress could induce increased SOD activity to protect *P. massoniana* from injury. With the aggravation of stress, a large number of free radicals were produced, which resulted in the destruction of the *P. massoniana* membrane system, aggravated membrane peroxidation and an increased MDA content. This is similar to the results of Huang et al. [21], Cui et al. [22] and Song et al. [23] for different plants. The balance of plant endogenous hormones plays an important role in plant adaptation to drought stress. ABA, a small but powerful hormone in plants, is an important signaling molecule that regulates the metabolic processes associated with stress adaptation and induces stress tolerance [24]. ABA content increased when plants were under drought stress and this led to stomatal closure and promoted the expression of many stress-related genes (such as Rab18 and RD29A), thus enhancing tolerance to water stress [25]. In this study, Gs decreased rapidly and ABA content increased significantly with increasing drought stress. It was considered that stomatal closure was correlated with ABA accumulation, and *P. massoniana* may regulate stomatal closure via ABA accumulation, thereby reducing the damage from osmotic stress and improving drought tolerance.

RNA sequencing (RNA-seq) is a new generation of genome sequencing (NGS) technology which has been an important means to detect the change in the gene transcription level in certain functional states of organisms. In this study, 162.04 gb clean data were obtained via transcriptome sequencing. Uni-genes were annotated using the public database and 35,398 single gene annotation results were obtained. In addition, to better understand the functionality of these DEGs, we made GO comments. The results showed that the GO classification included biological processes (25,976 genes), cellular components (19,839 genes) and molecular functions (26,525 genes), which were the most significant biological response processes under drought stress. Annotated genes were enriched in 131 KEGG pathways, which can be classified into five major metabolic pathways: plant hormone signal transduction, ribosomes, plant–pathogen interactions, RNA transport, plant circadian rhythm, and photosynthesis. The reliability of the transcriptome sequencing results was verified via qRT-PCR.

In response to drought stress, the increase in ABA content leads to the up-regulation of many downstream transcription factors and gene expression. We found that uni-genes in the “plant hormone signal transduction” pathway (ko04075) encoding PYR/PYL (Pm_118890), *ERF1* (Pm_62205) and AUX/IAA (Pm_107390) were significantly up- or down-regulated. The phosphorylation of target genes in plants and the activity of different structural proteins and TFs were regulated by MAPK. The activated TFs specifically bind to the cis-acting element of the corresponding target gene and initiate the transcription of the downstream specific response gene in response to drought stress [26]. *AP2/ERF* (APETALAP2/ethylen responsive factor) transcription factors are plant-specific transcription factors. GCAC (A/G)N(A/T)TCCC(A/G)ANG(C/T), GCC-box(AGCCGCC) and DRE/CRT(A/GCCCC), which specifically bind to the promoter region of stress-related genes, regulate gene expression. Yu et al. [27] found that the *OsERF109* gene combined with GCC-box/DRE/CRT elements negatively regulated rice drought resistance. According to the number and binding sequence of *AP2/ERF* transcription factor domains, *AP2/ERF* transcription factors were divided into five subfamilies, *AP2*, *ERF*, *DRE*, *RAV* and Soloist [28]. Studies have shown that the *ERF1* gene in *Arabidopsis thaliana* is involved in JA, EF and ABA signal transduction pathways, activating the expression of stress resistance genes, and *AtERF1*-overexpression. *Arabidopsis thaliana* significantly improves its resistance to drought, and transgenic plants can reduce leaf water loss through having a smaller stomatal pore size [8]. We found that under drought stress, the *ERF1* gene (*Pm_62205)* was significantly down-regulated, and plant hormone signal transduction was involved in functional annotation. In order to evaluate the role of the *PmERF1* transcription factor in drought tolerance, the *ERF1* gene with a high differential expression level was cloned, the pSuper:: *PmERF1* vector was transformed into wild-type *Populus davidiana* × *P. bolleana* via the leaf disc method, and then transgenic poplar plants over-expressing *PmERF1* were generated.

Water use efficiency (WUE) is an important physiological index with which to evaluate the drought resistance of plants. The WUE of plant leaves is directly proportional to photosynthesis and inversely proportional to transpiration [29]. Stomata play an important role in leaf tissue retention and gas exchange with the atmosphere. Stomatal closure is the first step for plants to retain water under drought stress, and about 90% of water loss (transpiration) occurs through stomata [30]. However, most research has focused on *Arabidopsis*, and very little has focused on woody plants. A reduction in Gs reduces the water requirements of plants [31]. Nevertheless, it also reduces photosynthesis and biomass accumulation [32]. In this study, the Gs and Tr of OE lines are lower than those of WT lines, and WT plants lost water faster than OE plants did (Figure 7F–H). It is possible that the decrease in stomatal conductance in overexpressed *PmERF1* plants may prevent water loss (Figure 7I), maintaining a high relative water content of leaves under drought stress (Figure 7E). Therefore, overexpressed *PmERF1* plants are more tolerant to drought stress.

## 4. Materials and Methods

### 4.1. Plant Material and Test Design

The half-sib family seed of *P. massoniana* was studied using the light matrix non-woven fabrics technique in 2017 at the National Forest Improved Seed Base of *P. massoniana* in Duyun City, Guizhou Province. A total of 200 pots were transplanted to Nanjing Forestry greenhouse in March 2018, with 1 plant per pot. The yellow soil (pH 5.0–5.5) was taken from the *P. massoniana* forest in the back hill of Nanjing Forestry University, and the soil content was controlled by weighing. The four groups of field moisture capacity were well-watered control (CK, 80 ± 5%), light drought (LD, 65 ± 5%), moderate drought (MD, 50 ± 5%) and severe drought (SD, 35 ± 5%). We further selected three seedlings with plenty of the stem apex needles from the each group as the samples for transcriptomic analyses. After 16 h of light culture (light intensity 35,000 lX) and 8 h of dark culture, the temperature was controlled at 15–22 °C, and the humidity was about 75%. The seedlings were sampled after a two-month period of drought treatment, and the stem apex needles of the seedlings were selected for RNA extraction. We also simultaneously collected the following tissues from 2-month-old *P. massoniana* seedlings: young leaf, senescent leaf, stem, and root. The samples were immediately frozen in liquid nitrogen.

Tissue culture seedlings of *Populus davidiana* × *P. bolleana* were cultured in our laboratory.

### 4.2. RNA Extraction, Library Construction and RNA-Seq

RNA was extracted from four treatment seedlings with three biological replicates for each treatment and then used to construct 12 cDNA libraries. Total RNA was extracted using RNAprep Pure Plant Kit (Tiangen, Beijing, China) following the manufacturer’s instructions. DNase I (Promega, Madison, WI, USA) was used to digest extracted RNA to remove DNA contamination. The purity and concentration of RNA was measured in accordance with the method of NanoDrop 2000 instrument (Thermo Fisher Scientific, Waltham, MA, USA). Purification of mRNA was carried out in accordance with the method described in Dynabeads mRNA Purification Kit (Invitrogen, Carlsbad, CA, USA). The cDNA libraries were constructed and sequenced using Illumina HiSeq2500 instrument by the Biomarker Technologies Co. (Ltd., Beijing, China, http://www.biomarker.com.cn/, accessed on 22 March 2023).

### 4.3. Analysis of Gene Expression and Differential Expression

The reads obtained from the sequence were compared with the FPKM (expected number of fragments per kilobase of transcript sequence per million base pairs sequenced) [33] values from Bowtie, and the expression level was estimated via RSEM. Differential gene expression analyses of different water conditions were analyzed using the R package (1.10.1) DESeq [34] (http://www.bioconductor.org/packages/release/bioc/html/DESeq2.html, accessed on 22 March 2023). The significant *p* values were corrected using the BH (Benjamini–Hochberg) method to ensure that the FDR was <0.05 and logFC was ≥2.

### 4.4. Feature Annotation and Enrichment Analysis

The clean data were assembled into uni-genes using Trinity sofware [35]. BlastX alignment between uni-genes and NR, Swiss-Prot, GO, COG, KOG, KEGG was performed. Gene function was annotated on the Pfam database [36] using HMMER. Coding sequences of uni-genes were predicted using TransDecoder. We imported the Nr results into the Blast2GO program and obtained all the selected genes with annotated GO terms. WEGO software was used to classify the GO terms. GO enrichment was performed using topGO. We assigned the assembled sequences using the KEGG pathway.

### 4.5. RNA Extraction and qRT-PCR Analysis

The tissue samples were ground into a fine powder in liquid nitrogen. Total RNA was isolated from all samples using RNAprep Pure Plant Plus Kit (DP441, Tiangen Biotech, Beijing, China) following the manufacturer’s instructions and stored at −80 °C. The 1st-strand cDNA synthesis kit (11141, Yeasen Biotech, Shanghai, China) was used to reverse transcribe RNA into cDNA, and the cDNA was stored at −20 °C.

In order to verify the reliability of the sequenced data, 9 genes differentially expressed to drought stress response were selected for quantitative real-time polymerase chain reaction (qRT-PCR) validation, and all gene-specific primers were designed using Primer 5.0 (Appendix A). SYBR Green (QPK-201, Toyobo Bio-Technology, Shanghai, China) reagents were used to detect the target sequence. Each PCR mixture (10 µL) contained 1 µL of diluted cDNA (20× dilution), 5 µL of SYBR Green Real-time PCR Master Mix, 0.4 µL of each primer (10 µM), and 3.2 µL of ddH2O. The PCR program had six stages: (1) 95 °C for 60 s (pre-incubation); (2) 95 °C for 15 s; (3) 60 °C for 15 s; and (4) 72 °C for 10 s, repeated 40 times (amplification); (5) 95 °C for 0.5 s; and (6) 60 °C for 1 min (melt). The PCR quality was estimated based on melting curves. TUA (α-tubulin) was used as the internal control [37]. Three independent biological replicates and three technical replicates for each biological replicate were examined. Quantification was successfully performed using comparative cycle threshold (Ct) values, and gene expression levels were calculated using the 2^−∆∆Ct^ method. The significance was determined via the *t*-test using SPSS statistical software (IBM, New York, NY, USA) (*p* < 0.05).

### 4.6. Subcellular Localization

*PmERF1* were selected for a transient expression experiment. The coding DNA sequence (CDS) regions were inserted into a pJIT166-GFP expression vector. Transient expression vectors (35S::*PmERF1*-GFP and 35S::*PmERF1*-GFP) containing green fluorescent protein (GFP) were transferred into the leaves of *N. benthamiana* following the method of Li [38]. The fluorescence signals were observed with an LSM 710 confocal microscope (Zeiss, Jena, Germany).

### 4.7. Plasmid Construction and Transformation

The *ERF1* ORF was combined into the pBI121 vector under the control of the 35S promoter by two restriction enzymes XbaI and SmaI. The construct was introduced into *A. tumefaciens* strain GV3101 and transformed into wild-type *Populus davidiana* × *P. bolleana* via the leaf disc method [39].

### 4.8. Molecular Verification

Genomic DNA was extracted from non-transgenic and transgenic lines using the CTAB method. Transformants were identified through PCR using the forward primer for the CAMV 35S promoter and the reverse primer for *ERF1* (Appendix A).

The transcript levels of *ERF1* in the transgenic poplar were confirmed using RT-qPCR. Total RNA was extracted from the leaves of WT and transgenic plants. First-strand cDNA was synthesized using one-step gDNA removal and cDNA Synthesis Kit (AT311, TransGen Biotech, Beijing, China). Primers were designed for RT-qPCR using Primer 5.0 (Appendix A).

### 4.9. Drought Experiments

Thirty WT and 60 OE (30 OE-2 and 30 OE-8) two-month-old plants were used as experimental materials. For the short-term drought experiment, plants in pots (140 cm in width and 125 cm in height) were subjected to drought by withholding watering for 7 days. Soil RWC, net CO_2_ assimilation, Gs, and transpiration were measured daily.

### 4.10. Determination of Photosynthetic Index, Physiological Characteristics and ABA Content

At 10:00 a.m. on a sunny day, the photosynthesis of *P. massoniana* seedlings and transgenic poplar were determined using a CIRAS-3 (PP Systems, Amesbury, MA, USA) photosynthesis meter, with 3 seedlings per treatment and 3 repeats.

The level of malondialdehyde (MDA) content in the needles (Ml) and roots (Mr) was determined using the thiobarbituric acid (TBA) method. Superoxide dismutase (SOD) activity was detected via the nitro-blue tetrazolium (NBT) method at 560 nm. ABA was detected using indirect competitive Elisa at 490 nm. All the kits used were purchased from Nanjing Jiancheng Bioengineering Institute (Nanjing, China) and the test was conducted in strict accordance with the instructions. Each indicator included 3 biological replicates (every 10 seedlings was mixed for 1 replicate).

## 5. Conclusions

In this study, we investigated the significant effects of sustained drought stress on physiological responses and gene expression in *P. massoniana*. Under continuous drought stress, SOD activity, MDA and ABA contents in coniferous leaves increased continuously, while photosynthesis decreased sharply. Transcriptomic analysis revealed that a large number of genes and transcription factors may participate in responding to drought stress, including genes regulating the ROS system, photosynthetic capacity, secondary metabolic biosynthesis and glucose metabolism, as well as *ERF*, *NAC*, *bHLH*, *MYB*, and other transcription factors. In this paper, *ERF1* genes with relatively high expression levels were screened from transcriptome data for cloning and the pSuper:: *PmERF1* vector was transformed into wild-type (WT) *Populus davidiana* × *P. bolleana* to assess the role of *PmERF1* in plants under drought stress. Under drought conditions, the over-expression of *PmERF1* could improve WUE via less transpiration and enhance the drought tolerance of plants under drought conditions. This study will contribute to better understanding the drought response mechanism of *P. massoniana* under stress.

## Figures and Tables

**Figure 1 ijms-24-11103-f001:**
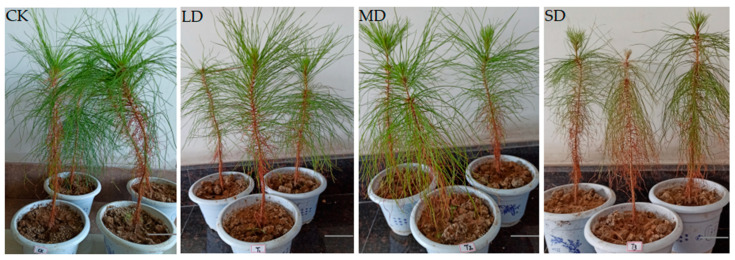
Phenotypes of *Pinus massoniana* seedlings under drought stress for 60 days. (CK) Normal water supply. (LD) Light drought stress. (MD) Moderate drought stress. (SD) Severe drought stress. Bars = 5 cm.

**Figure 2 ijms-24-11103-f002:**
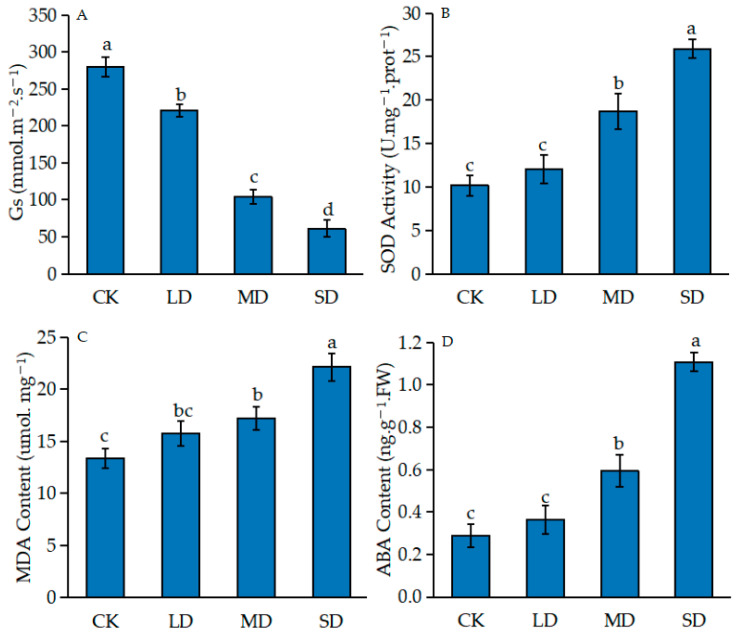
Physiological changes in *P. massoniansa* seedlings subjected to different drought stress conditions. (**A**) Stomatal conductance. (**B**) Superoxide dismutase. (**C**) Malonaldehyde. (**D**) Abscisic acid. Different lowercase letters above the bars represent significant differences (*p* < 0.05).

**Figure 3 ijms-24-11103-f003:**
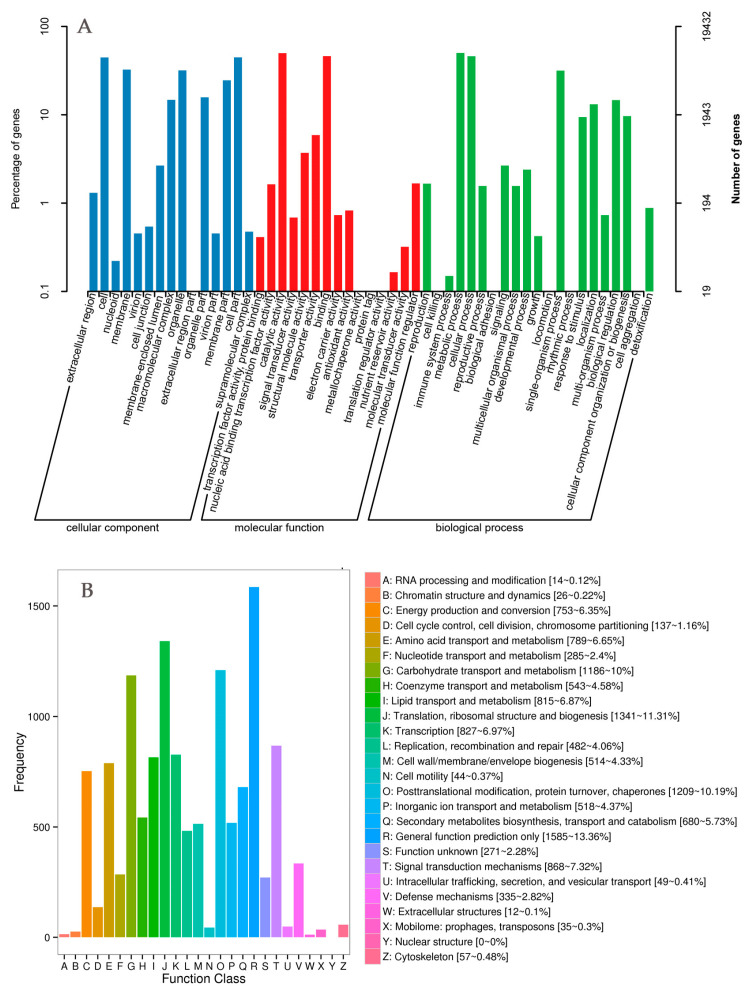
GO classification (**A**) and clusters of orthologous genes (**B**) of all identified genes.

**Figure 4 ijms-24-11103-f004:**
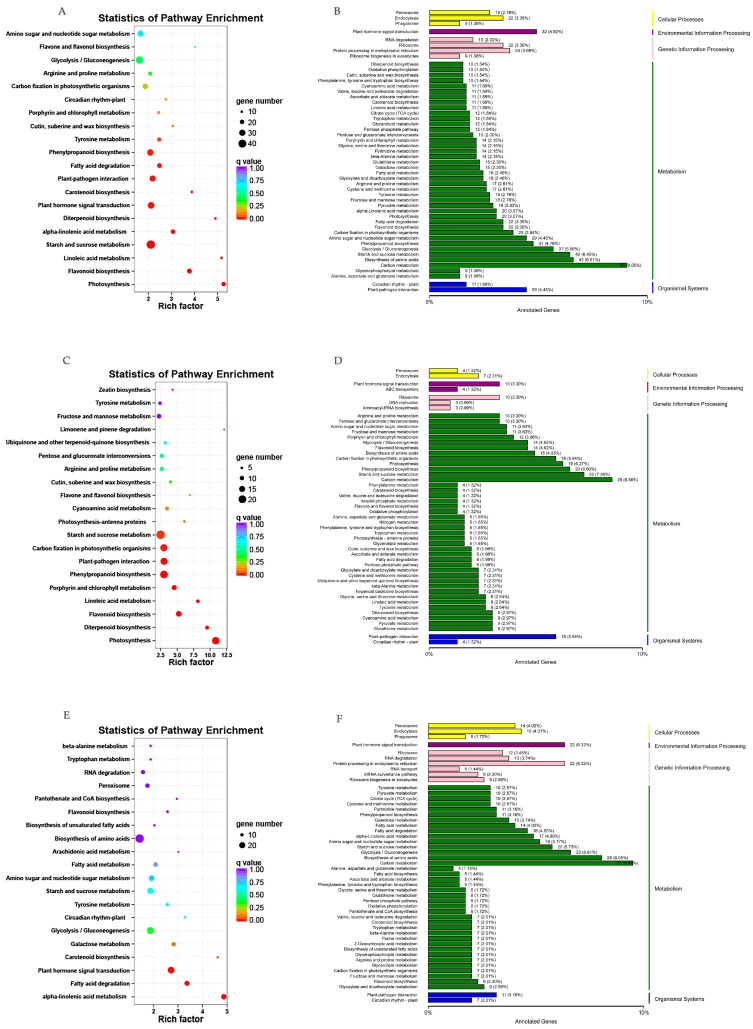
DEGs in drought-treated *P. massoniana*. (**A**,**B**) The KEGG pathway categories and enrichment factor analysis of SD vs. CK DEGs. (**C**,**D**) Up-regulated genes; (**E**,**F**) down-regulated genes.

**Figure 5 ijms-24-11103-f005:**
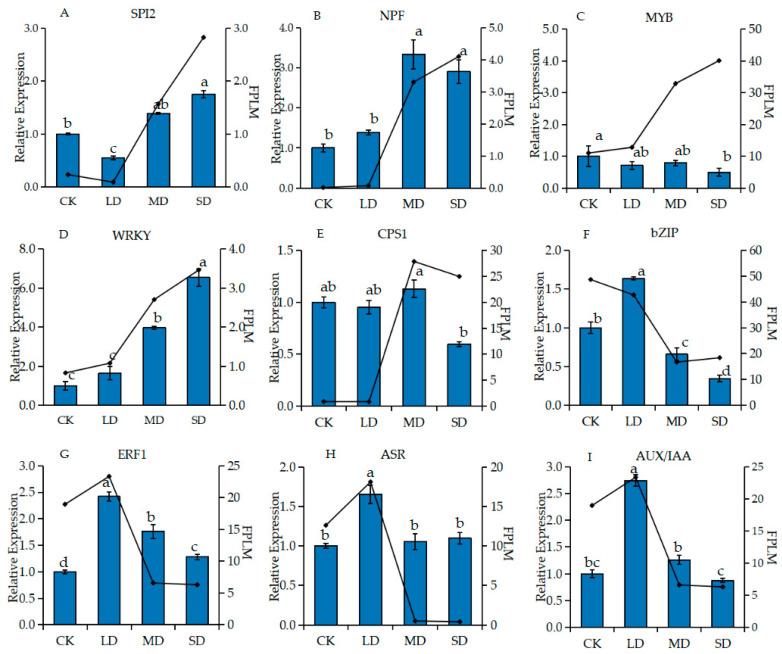
qRT-PCR validations of 9 candidate DEGs. Different lowercase letters above the bars represent significant differences (*p* < 0.05).

**Figure 6 ijms-24-11103-f006:**
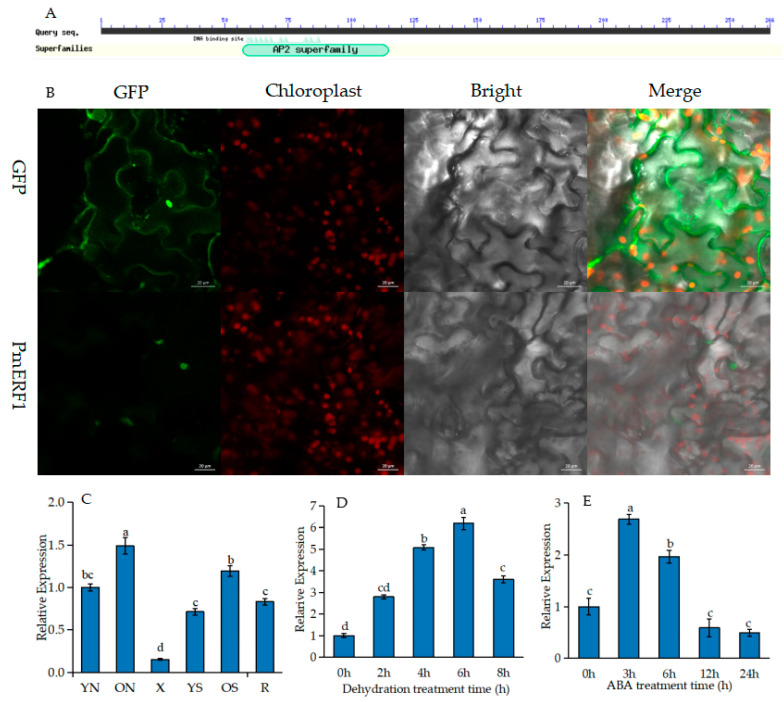
*ERF1* of *P. massoniana* (**A**) AP2 superfamily domain of *PmERF1*. (**B**) Subcellular localization of *PmERF1* protein. (**C**) Transcript levels of *PmERF1* in various tissues. YN, young needles; ON, old needles; X, xylem; YS, young stems; OS, old stems; R, roots. (**D**) *PmERF1* transcript levels under dehydration stress. (**E**) *PmERF1* transcript levels under ABA stress. Different lowercase letters above the bars represent significant differences (*p* < 0.05).

**Figure 7 ijms-24-11103-f007:**
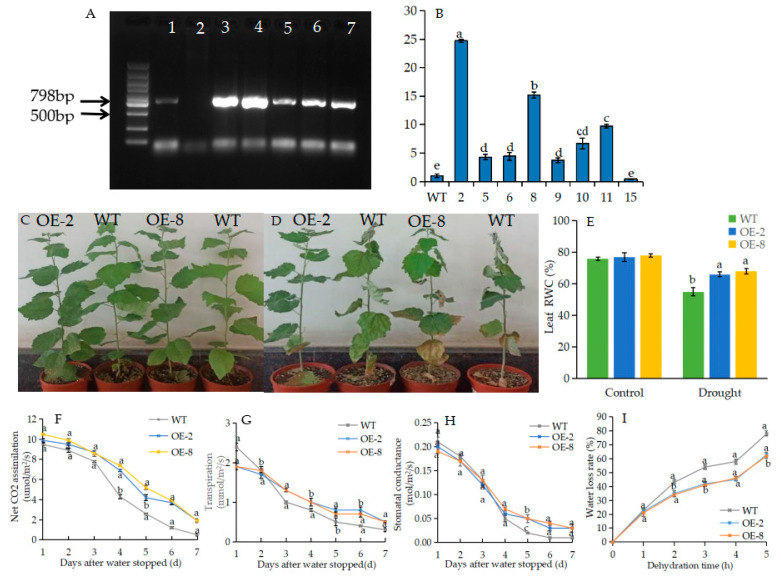
Overexpression of *PdbERF1* enhanced drought tolerance under drought conditions. (**A**) PCR confirmation of transgenic plants (1: the inserted target gene *PmERF1. 2*: Wild type. 3–7: Transgenic line). (**B**) qRT-PCR analysis of *PmERF1* expression level in different transgenic lines. The error bars represent the standard deviation of three biological replicates. (**C**) Before drought. (**D**) Drought for 7 d. (**E**) Leaf relative water content under non-stress and drought conditions. (**F**) A–drought duration curve. (**G**) Transpiration–drought duration curve. (**H**) Gs–drought duration curve. (**I**) Water loss from detached leaves. Means with different letters are significantly different at *p* < 0.05.

**Table 1 ijms-24-11103-t001:** Primers of nine uni-genes for quantitative PCR.

Gene Name	Gene_id	Gene Annotations	Gene Annotations
SPI2	c118913.graph_c0	Phenylpropanoid biosynthesis	Peroxidase
NPF	c126294.graph_c2	Amino acid transport and metabolism	Protein NRT1/PTR FAMILY
MYB	c123631.graph_c0	Myb-like DNA-binding domain	Transcription
WRKY	c126676.graph_c0	WRKY transcription factor 1	Sequence-specific DNA binding transcription factor activity
CPS1	c129306.graph_c0	Copalyl diphosphate synthase	Terpene synthase, N-terminal domain
bZIP	c118939.graph_c1	Basic region leucine zipper	Transcription factor HY5
ERF1	c62205.graph_c0	Plant hormone signal transduction	Ethylene-responsive transcription factor
ASR	c125386.graph_c0	ABA/WDS-induced protein	ASR protein
AUX/IAA	c107390.graph_c0	Function unknown	AUX/IAA family

## Data Availability

Not applicable.

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
