# Peer review of "The Transcriptomic Analysis of the Response of Pinus massoniana to Drought Stress and a Functional Study on the ERF1 Transcription Factor"

_ijms, 2023, doi:10.3390/ijms241311103_

Round 1
Reviewer 1 Report
- In this manuscript, a RNAseq experiment has been performed in Pinus massoniana in response to stress. The authors reported that the overexpression of AP2/ERF resulted in changes in response to drought stress. There are some important points that should be addressed before considering its publication in IJMS. I report bellow some of them.
I It is very confusing labelling as T1, T2 and T3 in Figures 1 and 2, and later in the manuscript is changed.
- The authors show activities referred to g-1 FW. It is more appropriate to use per mg of total protein (specific activities). Specially when the experiment is related to water stress and water content can change in the tissues.
- I found surprising the lack of statistically differences between T1 and T2 in Figure 2A. The same happens in other figures.
- It is confusing to have Table 1 after Figure 3. Table 1 is cited before in the text.
- The Figure 4 is missing in the manuscript.
- In legend to Figure 7, it is missing the relation to the inserted line.
- In legend to Figure 7, it is missing the presence of a second axe y in some figures.
- The point 2.5 Rt-qPCR analysis of drought-related genes. Should be detailed in the text.
Author Response
请参阅附件。

Reviewer 2 Report
This paper deals with the investigation of the drought response in Pinus massoniana. The authors performed RNAseq analysis and elucidated the molecular function of the gene encoding AP2/ERF-type transcription factor. Although the authors did many experiments, the presentation in the present paper is not appropriate. I could not see some parts of the figures, so I could not evaluate whether the contents are true or not.
1) In Figure 1, please add scale bars in the photos.
2) Although it might be problematic on my computer, I could not see Figure 4.
3) In Figure 6, I could not see the characters.
4) In Figure 7, the authors described that RNAseq and qPCR results are consistent. However, I do not think so. Some genes are consistent but others are not. Figure 7 suggests that RNAseq analysis are not reliable.
5) The authors focused on one gene encoding AP2/ERF transcription factor. However, there are no information regarding the gene including gene ID and homologues.
6) In Figure 8, I could not see the signal in nucleus. Where is the signal? If you say the signal is one small particle, you should show the positive controls of nucleus.
Reviewer 3 Report
This manuscript describes transcriptome analysis and functional analysis of ERF transcription factors in Pinus massoniana under drought stress; overexpression of an ERF transcription factor has been shown to enhance drought tolerance, contributing to a better understanding of drought stress responses in this plant. However, several issues should be addressed prior to acceptance of the paper.
1. The information on the homologs of the ERF transcription factors on which this paper focuses is only stated for the ERF of Pinus mugo. Phylogenetic analysis of the ERF transcription factors would be helpful to clarify which ERF they are closely related to and to assess whether the designation "ERF1" is appropriate.
2. The signal in the fluorescence microscopy image of PmERF1 in Fig. 8 is very weak. Is this observation done with the same exposure time and intensity as GFP? Why is the intensity of chloroplast autofluorescence different?
3. Figure 9A-D should be removed from the main figure and included in the supplemental figure if necessary.
4. The reviewer recommends that Figure 11 be revised or omitted. The involvement of PYR-PP2C-SnRK2-ABF in abscisic acid signaling has been demonstrated in various plants. This is not a major new finding in this paper. It would also be good to include PmERF1, the major new finding of this paper, in the signaling in Figure 11, if possible. Also, is PYR-PP2C-SnRK2-ABF only present in one gene in Pinus massoniana? If multiple homologs exist, they should be listed.
5. The RNA-seq data obtained in this paper should be deposited in a public database such as SRA in NCBI, and its accession number should be described. If the PmERF1 gene sequence was also newly identified in this paper, it should be deposited in Genbank, and the accession number should be listed.
6. The manuscript file is a track-change version. It should be plain text for the first submission.
7. line 224: RNA-sew -> RNA-seq
Reviewer 4 Report
Major comments
1] This manuscript has been just edited but not cleaned. It is a rough version. There are so many mistakes. Please make it neat & clean; carefully write & resubmit it.
2] Please submit transcriptome data in NCBI, then mention transcript id in manuscript.
3] Also mention gene accession number of all genes studied in RT-PCR data (Table 1).
4] Why did authors chose Poplar tree for drought resistant gene confirmation?
Minor comments
5] Mention reference for “Seamless cloning technology” “leaf disc method” in appropriate place in the manuscripts.
Reviewer 5 Report
The authors investigated the transcriptome of Pinus massoniana after drought treatment, and the function of ERF1 using transgenic plants. Revised manuscript is improved but still remains to be revised further.
1. There are so many main figures in the manuscript. Among the main figures, several figures are needed to move to supplementary figures. I suggest to move Figure 4, 5, 9, and 11 to supplementary figures.
2. What is the difference of the groups in between Figure 4B, and 4D. Are they the same grouping or not?
3. In Figure 8B, the exposure of pictures between GFP and pmERF1-GFP is different. Auto-flourescence intensity of chloroplast between GFP and pmERF1-GFP is different and intensity is very low in pmERF1-GFP compared to that in GFP. Adjust the exposure of pictures.
4. In Figure 10, the author have to demonstrate the generation of the transgenic plants. I could not find the information for the generation of transgenic plants..
5. There are typo in the label of x-axis (figure 10D and E). "water atopped"-> "water stopped"
Round 2
Reviewer 1 Report
The revised manuscript has been improved.
Reviewer 2 Report
I would like to confirm one minor point. In Figure 1, the scale bar length is 1 cm. Is this correct? If correct, the pot diameter is about 2 cm?
Author Response
According to your pointers, I have recalculated the scale bar length, the scale is 5cm, thank you for your instructions!

Reviewer 3 Report
The revised manuscript has addressed the majority of issues raised in a previous review.
Reviewer 4 Report
The manuscript is not properly written, not properly answered. Data not properly arranged.
